# Hemocyte differentiation to the megacyte lineage enhances mosquito immunity against *Plasmodium*

**Ana Beatriz F Barletta[1], Banhisikha Saha[1], Nathanie Trisnadi[1], Octavio AC Talyuli[1,2], Gianmarco Raddi[1], Carolina Barillas-Mury[1]\***

[1]Laboratory of Malaria and Vector Research, National Institute of Allergy and Infectious Diseases, National Institutes of Health, Rockville, United States; [2]Instituto de Bioquímica Médica Leopoldo de Meis, Universidade Federal do Rio de Janeiro, Rio de Janeiro, Brazil

**Abstract** Activation of *Toll* signaling in *Anopheles gambiae* by silencing *Cactus*, a suppressor of this pathway, enhances local release of hemocyte-derived microvesicles (HdMv), promoting activation of the mosquito complement-like system, which eliminates *Plasmodium* ookinetes. We uncovered the mechanism of this immune enhancement. *Cactus* silencing triggers a *Rel1*-mediated differentiation of granulocytes to the megacyte lineage, a new subpopulation of giant cells, resulting in a dramatic increase in the proportion of circulating megacytes. Megacytes are very plastic cells that are massively recruited to the basal midgut surface in response to *Plasmodium* infection. We show that *Toll* signaling modulates hemocyte differentiation and that megacyte recruitment to the midgut greatly enhances mosquito immunity against *Plasmodium*.

### Editor's evaluation

This article reveals a mechanism by which Toll pathway activation lead to protection against Plasmodium in Anopheles mosquitoes by promoting the differentiation of hemocytes.

**\*For correspondence:** cbarillas@niaid.nih.gov

## Introduction

Ookinete traversal of the *Anopheles gambiae* midgut disrupts the barriers that normally prevent bacteria of the gut microbiota from coming in direct contact with epithelial cells (*Kumar et al., 2010*), and this attracts hemocytes to the basal surface of the midgut (*Barletta et al., 2019*). *Plasmodium* ookinetes also cause irreversible damage to the cells they invade and trigger a strong caspase-mediated nitration response (*Han et al., 2000*; *Oliveira et al., 2012*, *Trisnadi and Barillas-Mury, 2020*). When hemocytes come in contact with a nitrated midgut surface, they undergo apoptosis and release hemocyte-derived microvesicles (HdMv) (*Castillo et al., 2017*). Local HdMv release promotes activation of thioester containing-protein 1 (*TEP1*) (*Castillo et al., 2017*), a major final effector of the mosquito complement-like system that binds to the parasite's surface and forms a complex that lyses the ookinete (*Blandin et al., 2004*).

Mosquito hemocytes are classified into three cell types, prohemocytes, oenocytoids, and granulocytes, based on their morphology. However, single-cell RNA sequencing (sc-RNAseq) analysis of *A. gambiae* hemocytes identified several novel subpopulations of granulocytes based on their transcriptional profiles and defined molecular markers specific for hemocyte subpopulations (*Raddi et al., 2020*). Furthermore, Lineage analysis revealed that regular granulocytes derive from prohemocytes

**eLife digest** Malaria causes hundreds of thousands of deaths each year. This devastating disease is caused by *Plasmodium* parasites, which are transmitted to people through female *Anopheles gambiae* mosquitos. Mosquitos become infected with *Plasmodium* when they ingest blood containing these malaria-causing parasites. However, *Plasmodium* must avoid the mosquito immune system to survive and spread.

The mosquito immune system is made up of several types of immune cells, including cells known as granulocytes. Granulocytes can further develop into additional cell subtypes, such as megacytes and antimicrobial granulocytes, but it is not clear how these types of cells work to protect mosquitos against infections.

In the mosquitos that transmit malaria, a cell signaling pathway called Toll helps control immune responses to disease-causing microbes, such as *Plasmodium*. When Toll signaling is strongly triggered in mosquitos, *Plasmodium* infection is eliminated because immune cell responses are enhanced – which results in lower levels of transmission to humans. But what is the underlying mechanism through which high levels of Toll signaling eradicate *Plasmodium* infection?

To find out, Barletta et al. collected cell samples from *A. gambiae* mosquitos and analyzed what happened when Toll signaling was strongly activated. They observed a large increase in the proportion of megacytes in these mosquitos (from 2% to 80% of all granulocytes). Toll signaling also caused megacytes to become bigger, cluster together, and have higher plasticity – meaning they could adopt different shapes. Barletta et al. used microscopy to show that these megacytes were releasing large mitochondria-like structures and membrane vesicles , which may be the trigger activating the mosquito's immune system. In live mosquitos, megacytes move towards the area of the *Plasmodium* infection and release microvesicles. These microvesicles are known to activate a part of the the mosquito's immune system called the complement-like system, destroying the parasites and preventing mosquito infection and disease transmission.

These findings show how strong Toll signaling triggers the mosquito immune system to eliminate *Plasmodium* infections. Understanding how the mosquito immune system tackles *Plasmodium* infection may help reveal ways to reduce or block transmission.

and can further differentiate into distinct cell types, including dividing granulocytes, and two final effector cells, megacytes and antimicrobial (AM) granulocytes (*Raddi et al., 2020*).

Silencing *Cactus*, a negative regulator of *Toll* signaling in *A. gambiae* mosquitoes, elicits a very strong *TEP1*-mediated immune response that eliminates *Plasmodium berghei* ookinetes (*Frolet et al., 2006*). This phenotype can be rescued by co-silencing *Cactus* with either *TEP1* or the *Rel1* transcription factor, indicating that parasite elimination is mediated by activation of *Toll* signaling, with *TEP1* as a final effector (*Frolet et al., 2006*). Later studies showed that hemocytes mediate this enhanced immune response, as transfer of *Cactus*-silenced hemocytes into naïve mosquitoes recapitulates the phenotype of systemic *Cactus* silencing (*Ramirez et al., 2014*). Furthermore, *Cactus* silencing also increases HdMv release in response to ookinete midgut invasion (*Castillo et al., 2017*), indicating that hemocytes are more reactive to *Plasmodium* infection. However, the nature of the functional changes in *Cactus*-silenced hemocytes that enhance immunity against *Plasmodium* are not known. Here, we explore the effect of *Cactus* silencing on circulating hemocyte populations and their response to infection of mosquitoes with bacteria and *Plasmodium*.

## Results

### Effect of *Cactus* silencing on mRNA markers of granulocyte populations

The effect of silencing *Cactus*, a suppressor of *Toll* signaling, on hemocyte differentiation was explored. Hemocytes that adhere to glass (mostly granulocytes) or that remain in suspension (mostly prohemocytes and oenocytoids) were collected 4 days post-injection from *dsLacZ* control and *dsCactus*-injected females. Bulk sequencing of cDNA libraries generated between 16.2 and 25.3 million fragments that mapped to the *A. gambiae* AgamP4.9 transcriptome. Only transcripts with 10 or more reads were included in the analysis, resulting in a total of 9421 unique transcripts (https://www.ebi.ac.uk/

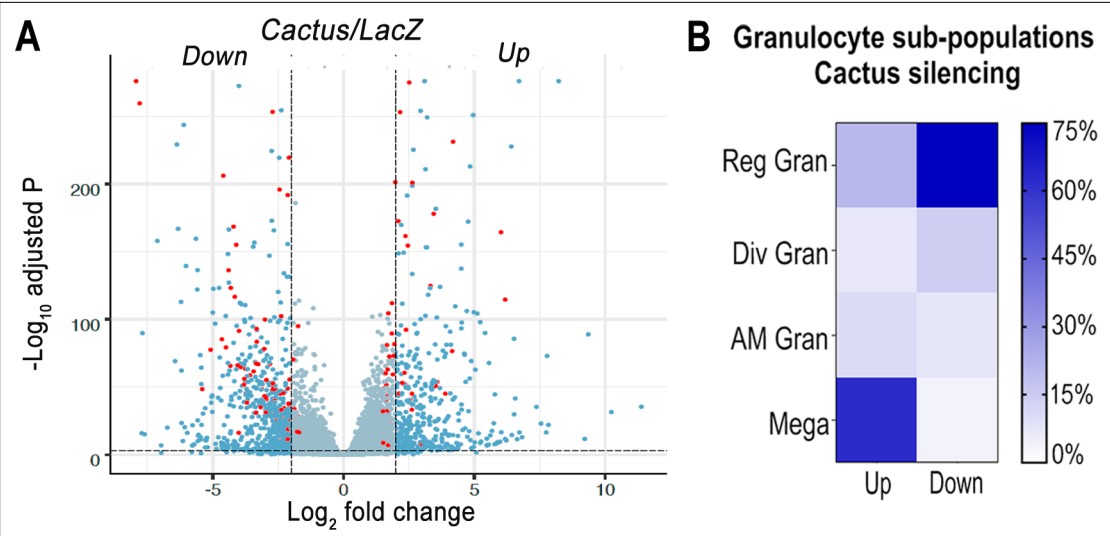

**Figure 1.** Effect of *Toll* pathway activation on mRNA markers of granulocyte populations. (**A**) Differential expression (DE) of *Cactus* dsRNA knockdown. From a total of 9421 filtered genes. Volcano plot of DE genes in *Cactus*-silenced hemocytes compared to *LacZ* control filtered for log2 fold change >2 and *Q* value <0.001. Dark blue dots on the right represent upregulated DE genes and on the left the downregulated ones. Red dots show genes that are hemocyte-specific markers. Complete list of up- and downregulated genes is listed in **Supplementary file 1** and **Supplementary file 2**. (**B**) Percentage of granulocyte subpopulation markers up- and downregulated in *Cactus*-silenced hemocytes. Complete list of up- and downregulated genes for each hemocyte subpopulation is in **Supplementary file 1** and **Supplementary file 2**. Reg Gran: regular granulocytes; Div Gran: dividing granulocytes; AM Gran: antimicrobial granulocytes; Mega: megacytes.

The online version of this article includes the following figure supplement(s) for figure 1:

**Figure supplement 1.** Quality control of *dsCactus* knockdown bulk RNAseq and differential expression (DE) between bound and unbound fractions.

). Glass-bound and unbound hemocyte samples were analyzed together, because the differences in expression between ds*LacZ* versus ds*Cactus*-silenced hemocytes explained 81% of the variance between the four experimental groups (*Figure 1—figure supplement 1A and B*). Differential expression (DE) analysis of *Cactus*-silenced hemocytes using the DESeq2 software identified 1071 differentially expressed genes (*Q* value <0.001), of which 407 were upregulated (log2 fold change >2), while 664 were downregulated (log2 fold change <−2) (*Figure 1A*).

The effect of *Cactus* silencing on expression of the transcripts that define the different hemocyte clusters established by sc-RNAseq (*Raddi et al., 2020*) was analyzed (*Supplementary file 1* and *Supplementary file 2*), to establish whether there was a significant effect on the relative abundance of specific hemocyte subpopulations. Overall, 23 oenocytoid markers, 2 from prohemocytes, and 57 from granulocytes were differentially expressed between ds*LacZ* and ds*Cactus* hemocytes (*Supplementary file 1* and *Supplementary file 2*). Most differentially expressed oenocytoid markers 22/23 (95%) were downregulated, while one of the prohemocyte markers was upregulated and the other one was downregulated (*Figure 1B*). The number of downregulated granulocyte markers 28/57 (49%) was very similar to that of upregulated ones 29/57 (51%). However, detailed analysis of granulocyte subpopulations revealed that most upregulated markers 18/29 (62%) correspond to megacytes, while most downregulated markers correspond to regular granulocytes 21/28 (75%). This suggests that ds*Cactus* silencing increases the proportion of circulating megacytes, at the expense of a reduction in regular granulocytes. As expected, expression of several genes involved in *Toll* signaling or final effector this pathway, such as *Toll*-like receptors, *CLIP* proteases, Serpins, C-type Lectins, and *Defensin* are upregulated in ds*Cactus* hemocytes (*Supplementary file 3*).

## *Cactus* silencing promotes granulocyte differentiation into megacytes

*Cactus* silencing did not significantly increase the proportion of total circulating granulocytes, based on hemocyte counts by light microscopy (*Figure 2—figure supplement 1*), suggesting that the observed enhanced immune response could be due to functional changes in hemocytes. The morphology of hemocytes perfused from *Cactus*-silenced females was analyzed using fluorescent probes to stain the

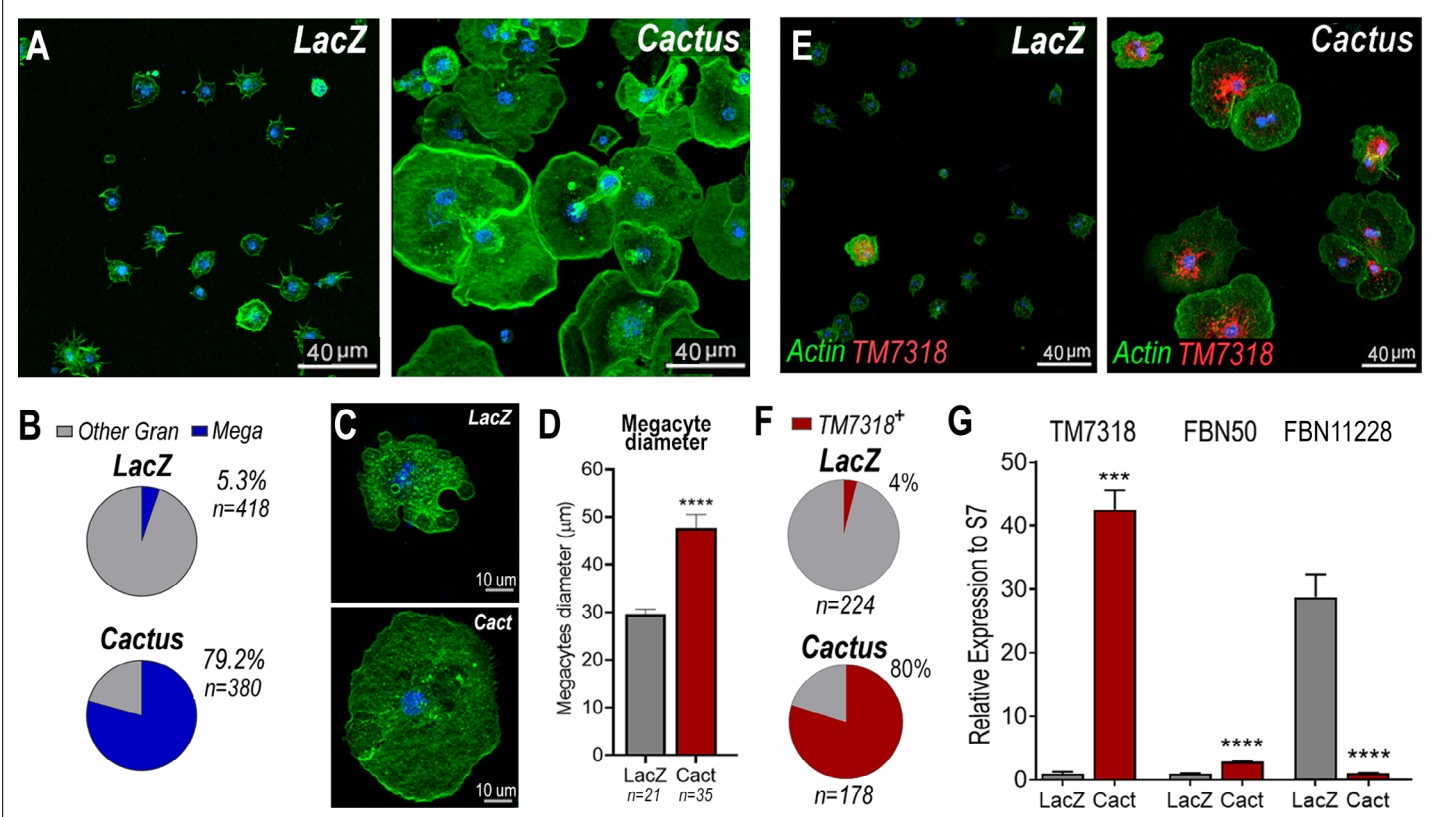

**Figure 2.** *Cactus* silencing promotes granulocyte differentiation into megacytes. (**A**) *A. gambiae* hemocytes in LacZ control and *Cactus* attached to a glass surface. Actin is shown in green and nuclei in blue. Scale bar: 40 µm. (**B**) Percentage of megacytes among all granulocytes in *dsLacZ* and *dsCactus* mosquitoes. Percentages were compared using $X^2$ test. **** = p ≤ 0.0001. (**C**) Megacyte in control *LacZ* mosquitoes (upper) and in *Cactus*-silenced mosquitoes (lower). Actin is showing in green, and nuclei is in blue. Scale bar: 10 µm. (**D**) Diameter of megacytes from *LacZ* control and *Cactus*-silenced mosquitoes. Error bars represent mean ± standard error of the mean (SEM). Unpaired *T*-test. **** = p ≤ 0.0001. (**E**) RNA in situ hybridization for megacyte-specific marker *TM7318*. Actin is shown in green (phalloidin), *TM7318* mRNA in red, and the nuclei in blue (Hoechst). Scale bar: 40 µm. (**F**) Percentage of *TM7318*-positive cells in *LacZ* and *Cactus*-silenced granulocytes. Percentages were compared using $X^2$ test. **** = p ≤ 0.0001. (**G**) Relative mRNA expression of hemocyte-specific markers in *LacZ* control and *Cactus* hemocytes for transcriptome validation. Megacyte marker (*TM7318*), antimicrobial granulocytes (*FBN50*), and regular granulocytes (FBN11228). Gene expression was normalized using *RpS7* expression. Error bars represent mean ± SEM. Unpaired *T*-test, ***=p ≤0.001, = p ≤ 0.0001. Two independent experiments with 3 replicates each experiment, a total of 6 pools of 15 mosquitoes each.

The online version of this article includes the following figure supplement(s) for figure 2:

**Figure supplement 1.** Megacyte increase in response to *Cactus* silencing is not a result of hemocyte proliferation.

**Figure supplement 2.** *Toll* activation controls megacyte differentiation and not proliferation.

actin cytoskeleton and the nucleus. *Cactus* silencing dramatically increased the proportion of large granulocytes (diameter >40 µm), presumably megacytes, from 5.3% to 79.2% (p < 0.0001, $X^2$ test) (*Figure 2A, B*), in agreement with the observed increase in upregulated megacyte-specific markers in the transcriptomic analysis of *Cactus*-silenced hemocytes (*Figure 1B*). Interestingly, megacytes from *Cactus*-silenced mosquitoes (*Figure 2A*) are even larger (average diameter of 47 µm after spreading in a glass surface) than megacytes from *dsLacZ* controls (average diameter of 30 µm) (*Figure 2C, D*). In situ RNA hybridization of *dsCactus* granulocytes with a fluorescent probe for the megacyte-specific marker *TM7318*, confirmed that the proportion of *TM7318*-positive granulocytes was much higher (80%) in *Cactus*-silenced females than in *dsLacZ* controls (4%) (p < 0.0001, $X^2$ test) (*Figure 2E, F*), providing direct evidence that overactivation of *Toll* signaling triggers a dramatic increase in the proportion of circulating megacytes. Expression analysis of the *TM7318* marker in perfused hemocyte samples confirmed that mRNA levels were 42-fold higher in *dsCactus* hemocytes than the *dsLacZ* control group (p < 0.001, *T*-test) (*Figure 2G*), while a modest increase (2.8-fold) in *FBN50* mRNA, a marker of AM effector granulocytes, was observed (p < 0.0001, *T*-test). Conversely, expression

of *FBN11228*, a marker of regular granulocytes, decreased by 30-fold in circulating hemocytes of *Cactus*-silenced mosquitoes (*Figure 2G*). The changes in the relative abundance of mRNAs from cell-specific markers in *dsCactus* hemocytes agrees with the observed changes in hemocyte morphology and the in situ hybridization (ISH) and transcriptomic data (*Figure 2G*).

The relative increase in megacytes in *Cactus*-silenced *A. gambiae* females could be due to enhanced megacyte proliferation or to increased differentiation of regular granulocytes into megacytes. A total hemocyte count revealed that the number of circulating hemocytes was not significantly different between *dsLacZ* (mean 17,151/mosquito, *n* = 14) and *dsCactus* mosquitoes (mean 23,477/mosquito, *n* = 14, Mann–Whitney *T*-test) (*Figure 2—figure supplement 1A*) 2 days post-injection. The mean number of total hemocytes was lower 4 days post-silencing, but there was also no significant difference between *dsLacZ* (mean 8725/mosquito, *n* = 14) and *dsCactus* mosquitoes (mean 10,836/mosquito, *n* = 14, Mann–Whitney *T*-test) (*Figure 2—figure supplement 1A*). Furthermore, there was no difference in the proportion granulocytes at 2 days, *dsLacZ* (3.62%) and *dsCactus* females (3.48%, Mann–Whitney *T*-test), and 4 days, *dsLacZ* (3.91%) and *dsCactus* females (5.47%, Mann–Whitney *T*-test), post-injection. In contrast, a significant increase in the proportion of megacyte was already apparent 2 days post-injection in *Cactus*-silenced mosquitoes (2.08% of all hemocytes, p < 0.0001, Mann–Whitney *T*-test), relative to *dsLacZ* controls (0.07%) (*Figure 2—figure supplement 1B*); with a corresponding decrease in the proportion of other granulocytes from 3.5% in *dsLacZ* controls to 1.4% in *dsCactus* (*Figure 2—figure supplement 1B*). At 4 days post-injection, the differences were more pronounced, with the proportion of

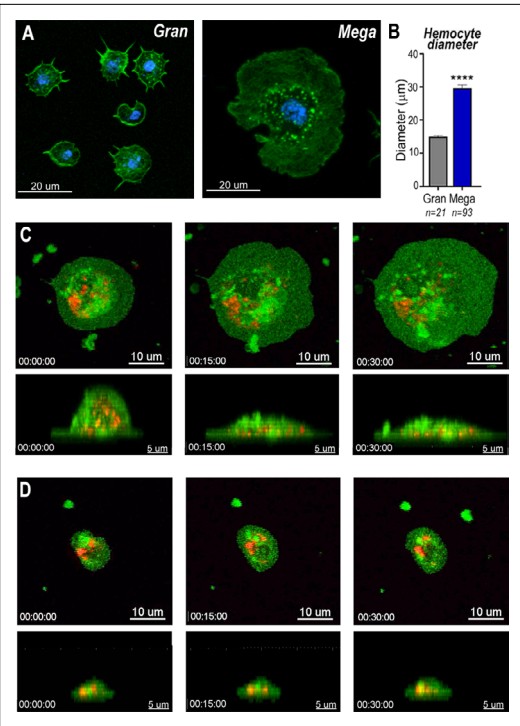

**Figure 3.** Snapshots of megacyte and granulocyte cell dynamics. (**A**) Regular granulocytes and megacytes from *A. gambiae* females spread on a glass surface. Actin, green (phalloidin) and nuclei, blue (Hoechst). Scale bar: 20 μm. (**B**) Granulocyte diameter of sugar-fed mosquitoes after spreading on a glass surface. Error bars represent mean ± standard error of the mean (SEM). Unpaired *T*-test. ****p ≤ 0.0001. (**C**) Live imaging time-lapse of a megacyte spreading in a glass surface for 30 min. Plasma membrane stained in green and microvesicles in red. Top (XY) and lateral view (XZ) of a megacyte. Scale bars: 10 and 5 μm, respectively. (**D**) Live imaging time-lapse of a granulocyte spreading on a glass surface for 30 min. Top (XY) and lateral view (XZ) of a regular granulocyte. Scale bars: 10 and 5 μm, respectively (see *Videos 1–4*).

megacytes reaching 3.7% in *Cactus*-silenced mosquitoes (p < 0.0001, Mann–Whitney *T*-test) relative to *dsLacZ* controls (0.07%) (*Figure 2—figure supplement 1C*), with a corresponding decrease in other granulocytes from 3.8% in *dsLacZ* to 1.8% in *dsCactus* mosquitoes (*Figure 2—figure supplement 1C*). Taken together, these data indicate that, although the total number of hemocytes and the percentage of total granulocytes remained unchanged in response to *dsCactus* silencing, the proportion of megacytes increased at the expense of other granulocytes.

The effect of *Cactus* silencing on granulocyte proliferation was evaluated by quantitating the proportion of hemocytes that incorporated Bromodeoxyuridine/5-bromo-2'-deoxyuridine (BrdU), a thymidine analog. The proportion of BrdU+ hemocytes that adhered to glass (mostly granulocytes) in *dsCactus* mosquitoes (51%, *n* = 694 cells) is not significantly different from dsLacZ controls (52%, *n* = 410 cells) (*Figure 2—figure supplement 2A, B*). BrdU fluorescence intensity (RFU) is also not significantly different between *dsLacZ* and *dsCactus* hemocytes (*Figure 2—figure supplement 2C*). However, the ratio of BrdU fluorescence intensity to nuclear volume is significantly lower in *dsCactus* hemocytes (*Figure 2—figure supplement 2D*). This indicates that the increase in nuclear volume in megacytes does not involve DNA replication. However, BrdU labeling can be lost over time,

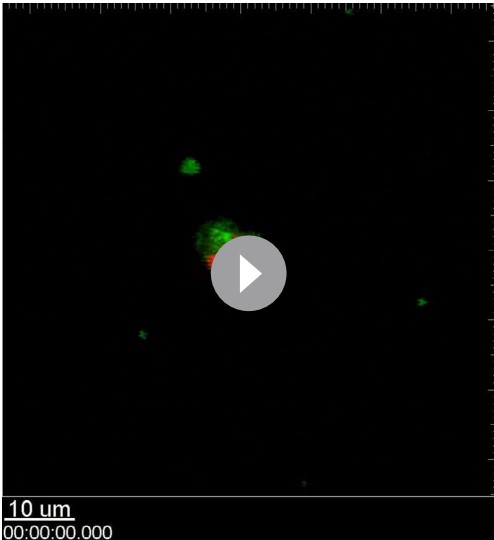

**Video 1.** Top view (XY) of a regular granulocyte from *A. gambiae* mosquito female. Showing in red is the microvesicle staining and in green the plasma membrane. Scale bar: 10 μm. Hemocyte was imaged for 1 hr in intervals of 5 min.

https://elifesciences.org/articles/81116/figures#video1

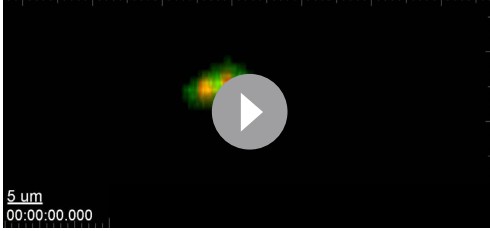

**Video 2.** Side view (XZ) of a regular granulocyte from *A. gambiae* mosquito female. Showing in red is the microvesicle staining and in green the plasma membrane. Scale bar: 5 μm. Hemocyte was imaged for 1 hr in intervals of 5 min.

https://elifesciences.org/articles/81116/figures#video2

making it hard to establish when DNA replication occurred. The proportion of hemocytes undergoing mitosis after *Cactus* silencing was directly evaluated using phospho-Histone H3 (PHH3) staining, which only labels mitotically active cells. Two days post-injection, the proportion of PHH3+ hemocytes that adhered to glass (mostly granulocytes) in *dsCactus* mosquitoes (0.7%, *n* = 896) was small, and not significantly different from *dsLacZ* controls (0.6%, *n* = 718 cells) (*Figure 2—figure supplement 1D*). At 4 days the proportions were also similar, with very few hemocytes positive for PHH3 staining both in *dsLacZ* (0.5%, *n* = 799 cells) and *dsCactus* (0.3%, *n* = 1039) (*Figure 2—figure supplement 1D*). Moreover, the few hemocytes that were positive for PHH3 in *dsCactus* mosquitoes did not have the characteristic size or morphology of megacytes (*Figure 2—figure supplement 1D*). These observations, together with the increase in the proportion of megacytes in *dsCactus* females, at the expense of other regular granulocytes (*Figures 1B and 2G*), indicate that *Cactus* silencing promotes differentiation of granulocytes to the megacyte lineage.

## Cell dynamics of mosquito granulocytes

Megacytes are about twice as large as regular granulocytes. Regular granulocytes (*Figure 3A*) reach an average diameter of 14.2 μm (*Figure 3B*) when they spread over a glass surface, while the average diameter of megacytes is 28.6 μm (p < 0.0001, unpaired *T*-test) (*Figure 3A, B*). Granulocyte cellular dynamics was evaluated by live imaging of perfused hemocytes in vitro as they adhered and spread on a glass surface. Hemocytes were labeled in vivo, through systemic injection of adult females with a red lipophilic dye (Vybrant CM-DiI) that accumulates on intracellular vesicles. After perfusion, a green, fluorescent probe (Cell Mask) was added to label the plasma membrane. Both regular granulocytes and megacytes attached to the glass surface and spread fully within 1 hr (*Videos 1–4*). Megacytes already have a larger cell diameter when they first attach to glass (*Figure 3C*, upper panel and *Video 3*),

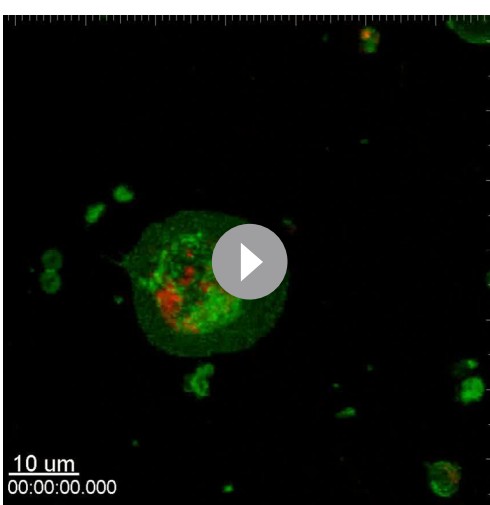

**Video 3.** Top view (XY) of a megacyte from *A. gambiae* mosquito female. Showing in red is the microvesicle staining and in green the plasma membrane. Scale bar: 10 μm. Hemocyte was imaged for 1 hr in intervals of 5 min.

https://elifesciences.org/articles/81116/figures#video3

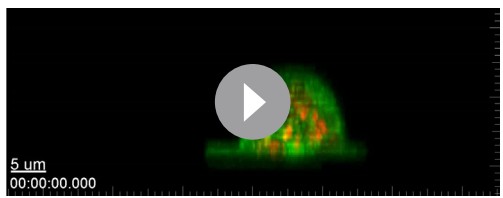

**Video 4.** Side view (XZ) of a megacyte from *A. gambiae* mosquito females. Showing in red is the microvesicle staining and in green the plasma membrane. Scale bar: 5 μm. Hemocyte was imaged for 1 hr in intervals of 5 min.

https://elifesciences.org/articles/81116/figures#video4

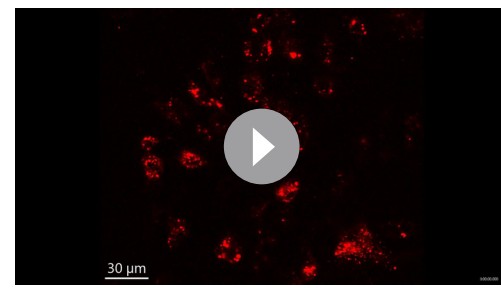

**Video 6.** In vivo hemocyte patrolling activity in *dsCactus* mosquitoes. Hemocytes stained in red were imaged through the cuticle of the mosquito for 1 hr and 20 min. Scale bar: 30 μm.

https://elifesciences.org/articles/81116/figures#video6

and exhibit a peripheral 'halo', corresponding to an area of extended thin cytoplasm, almost devoid of vesicles (*Figure 3C*, upper panel and *Video 3*). Lateral views revealed that, initially, megacytes have a large nucleus and a voluminous cytoplasm in the central region of the cell that flattens dramatically as the cell 'spreads' over the glass surface (*Figure 3C*, lower panel and *Video 4*). In contrast, the central region of regular granulocytes remains mostly unchanged (*Figure 3D*, lower panel and *Video 2*) and the periphery of the cell exhibits a modest increase in diameter as the cell spreads along the surface (*Figure 3D*, upper panel and *Video 1*).

## Characterization of megacyte in vivo dynamics and ultrastructure

The effect of *Cactus* silencing on granulocyte dynamics was evaluated in vivo, through live imaging of hemocytes circulating in adult female mosquitoes. Female mosquitoes were imaged for 2 hr, 1 day after blood feeding on a healthy mouse. Hemocytes were visualized by systemic injection of Vybrant CM-DiI, a fluorescent lipophilic dye that is preferentially taken up by granulocytes. Circulating hemocytes in *dsLacZ* females (presumably normal granulocytes) have a smaller diameter than those of *dsCactus* females (*Videos 5 and 6*; *Figure 3A, B*), and they seldom come in contact with each other as they patrol the basal surface of the midgut (*Video 5*). Hemocytes from *dsCactus* females (presumably megacytes) are larger and have a spindle shape (*Figure 3A, B*; *Videos 3 and 4*). They appear to have higher plasticity, as they can readily stretch their cytoplasm and often come into contact with each other (*Video 6*). The plasticity of *dsCactus* megacytes was confirmed by in vitro live imaging of perfused hemocytes labeled by systemic injection of Vybrant CM-DiI and green Cell Mask. Some megacytes from *dsCactus* mosquitoes projected long thin filopodia toward other megacytes (*Video 7*). This process was not observed in regular granulocytes or in megacytes

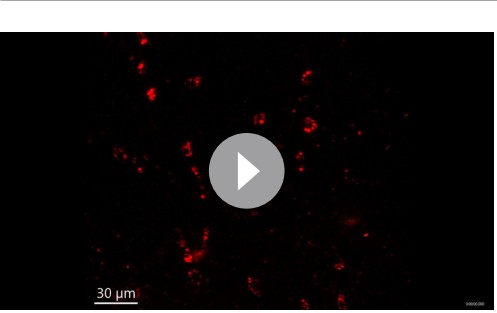

**Video 5.** In vivo hemocyte patrolling activity in *dsLacZ* mosquitoes. Hemocytes stained in red were imaged through the cuticle of the mosquito for 1 hr and 20 min. Scale bar: 30 μm.

https://elifesciences.org/articles/81116/figures#video5

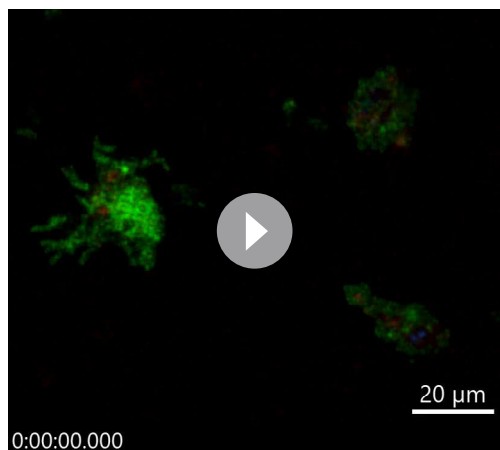

**Video 7.** In vitro dynamics of *dsCactus* megacytes. Perfused hemocytes from *dsCactus* mosquitoes. Plasma membrane is showing in green, microvesicles in red, and nuclei in blue. Scale bar: 20 μm.

https://elifesciences.org/articles/81116/figures#video7

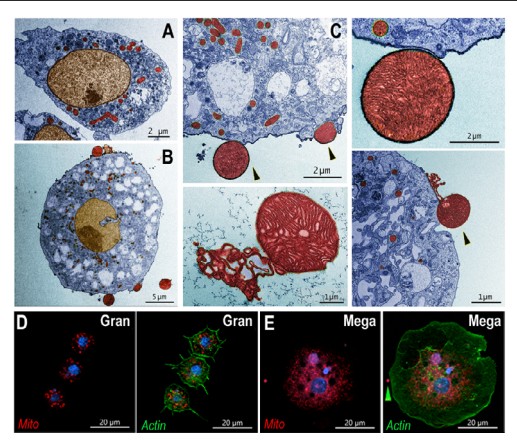

**Figure 4.** Ultrastructure of megacytes in *Cactus*-silenced mosquitoes. (**A**) Transmission electron microscopy (TEM) of regular granulocytes from *Cactus*-silenced mosquitoes. Scale bar: 2 μm. (**B**) TEM of megacytes from *Cactus*-silenced mosquitoes. Scale bar: 5 μm. (**C**) Extracellular giant mitochondria-like structures (black arrows). Closeup of a mitochondria-like structure (lower center). Scale bars: 2 and 1 μm. TEM images were digitally colorized, cytoplasm is shown in blue, mitochondria in red, and nuclei in golden yellow. (**D**) Mitotracker staining in regular granulocytes. Scale bar: 20 μm. (**E**) Mitochondrial staining of *Cactus*-silenced megacytes. Actin is stained in green (phalloidin), mitochondria in red (mitotracker), and nuclei in blue (Hoechst). Yellow arrow indicates an extracellular mitochondrion like structure outside of a megacyte. Scale bar: 20 μm.

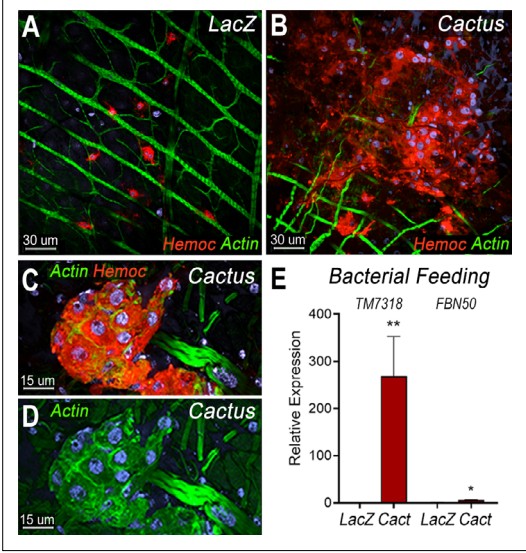

**Figure 5.** Bacterial feeding increases megacyte association to the midgut basal surface. (**A**) Effect of bacterial feeding in *LacZ*-injected controls on hemocytes associated to the midgut basal surface. (**B**) Effect of *Cactus* silencing on the hemocytes associated to the basal surface of the midgut 4 hr post-bacterial feeding. (**A, B**) Scale bar: 30 μm. (**C, D**) Hemocyte cluster attached to the midgut surface in *Cactus*-silenced mosquitoes 4 hr post-bacterial feeding. Scale bar: 15 μm. (**A–D**) Midgut actin is shown in green (phalloidin), hemocytes (stained with Vybrant CM-DiI) in red, and nuclei in blue (Hoechst). (**E**) Relative mRNA levels of effector hemocyte markers in the midgut 4 hr after bacterial feeding in *LacZ* and *Cactus*-silenced mosquitoes. Scale bar: 15 μm. Error bars in (**E**) represent mean ± standard error of the mean (SEM). Unpaired *T*-test, *p ≤ 0.05, **p ≤ 0.01. Two independent experiments with 3 replicates for each experiment, a total of 6 pools of 15 mosquitoes each.

from the *dsLacZ* controls. Taken together, our live imaging data indicate that, in addition to their larger diameter (*Figure 2C, D*), *dsCactus* megacytes are also more active, have increased plasticity as they patrol the midgut (*Video 6*), and greater tendency to interact with each other and form clusters (*Videos 6 and 7*).

The detailed ultrastructure of megacytes was explored using transmission electron microscopy (TEM). Hemocytes from *Cactus*-silenced females were collected by perfusion, fixed in suspension, and allowed to settle. As expected, the maximum diameter of hemocytes fixed while in suspension was smaller than when they were allowed to spread on a glass surface. However, regular granulocytes were still significantly smaller (6–10 μm) than megacytes (15–20 μm), with nuclei that are also proportionally smaller (*Figure 4A, B*). Extensive electrodense areas are observed in the nuclei of megacytes, probably corresponding to the nucleolus. Large numbers of cytoplasmic vacuoles that contain abundant amorphous material are observed, as well as an extensive mitochondrial network (*Figure 4A, B*). Mitochondrial organization of perfused hemocytes was further investigated using Mitotracker staining. Mitochondria of regular granulocytes have a punctate pattern with strong staining on individual organelles (*Figure 4D*). In contrast, megacytes exhibit a more diffuse and extensive mitochondrial network (*Figure 4E*). It is noteworthy that large membrane-bound mitochondria-like extracellular structures and small vesicles are often observed 'budding off' from the surface of *dsCactus* megacytes (*Figure 4B, C*), but not from regular granulocytes (*Figure 4A*).

## Megacytes associate with the basal surface of the midgut in response to bacterial feeding

We have shown that direct contact of bacteria with epithelial cells, before the peritrophic matrix is formed, triggers PGE2 release and attracts hemocytes to the basal surface of the midgut (*Barletta et al., 2019*). Hemocyte recruitment to the midgut in *dsCactus* females was explored by providing a bovine serum albumin (BSA) protein meal containing bacteria. As expected, bacterial feeding attracted hemocytes to the midgut surface in both *dsCactus* and *dsLacZ* control females (*Figure 5A, B*). However, there are important differences in hemocyte recruitment between them. In *dsLacZ* females, hemocytes attach to the midgut basal lamina individually or in doublets (*Figure 5A*), while hemocytes from *dsCactus* females form large clusters on the basal midgut surface, with multiple hemocytes in very close association (*Figure 5B, C*). *dsCactus* hemocytes on the midgut surface have the characteristic morphology of megacytes, with a larger cytoplasm and nuclei than those from *dsLacZ* hemocytes (*Figure 5B, C*). Accumulation of actin was often observed in the boundaries where hemocytes from *dsCactus* females come in direct contact with each other as they form extensive clusters (*Figure 5D*).

The recruitment of granulocyte subpopulations to the midgut of *dsCactus* females in response bacterial feeding was confirmed by quantitation of midgut-associated mRNAs transcripts of markers expressed in specific hemocyte subpopulations. *TM7318* mRNA levels increased dramatically in *dsCactus* midguts after bacterial feeding (250-fold increase) relative to *dsLacZ* control (p = 0.0022, Mann–Whitney test) (*Figure 5E*), indicative of extensive megacyte recruitment. A significant, but more modest increase in *FBN50* (fivefold) (p = 0.0152, Mann–Whitney test) a marker of AM granulocytes, was also observed (*Figure 5E*).

## *Toll* signaling is required for megacyte differentiation and *Plasmodium* ookinete elimination in *Cactus*-silenced females

To establish whether differentiation of granulocytes to the megacyte lineage in *dsCactus* females was mediated by the *Toll* pathway, the effect of co-silencing the *Rel1* transcription factor was evaluated. As expected, *Cactus* silencing dramatically increased the proportion of megacytes, from 3.5% to 76% (p < 0.0001, $X^2$ test) (*Figure 6A, B*). Co-silencing *Cactus* and *Rel1* reverted this effect, resulting in a proportion of megacytes (6.6%) not significantly different from *LacZ* controls

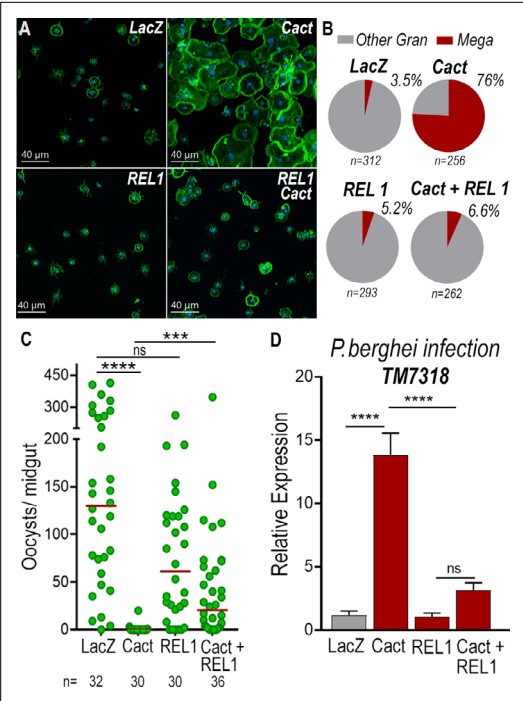

**Figure 6.** *Toll* signaling is required for megacyte differentiation and *Plasmodium* ookinete elimination in *dsCactus* females. (**A**) *A. gambiae* hemocytes in *LacZ* control, *Cactus*, *Rel1*, and *Cactus* + *Rel1* attached to a glass surface. Actin is shown in green and nuclei in blue. Scale bar: 40 µm. (**B**) Percentage of megacytes among all granulocytes in *dsLacZ*, *dsCactus*, ds *Rel1*, and *dsCactus* + *Rel1* mosquitoes. Percentages were compared using $X^2$ test. ****p ≤ 0.0001. (**C**) Mosquito susceptibility to *P. berghei* infection after dsRNA injection for *LacZ*, *Cactus*, *Rel1*, and *Cactus* + *Rel1*. Each dot in C represents the number of oocysts or hemocytes, respectively, for individual midguts. The median is indicated by the red line. Analysis of variance (ANOVA) Kruskal–Wallis test, ****p ≤ 0.0001; ***p ≤ 0.001, NS, p = 0.1074. (**D**) Relative mRNA levels of *TM7318*, megacyte marker, in the midgut 26 hr post-*P. berghei* infection (post-invasion) in *LacZ*, *Cactus*, *Rel1*, and *Cactus* + *Rel1* silenced mosquitoes. Error bars in (**D**) represent mean ± standard error of the mean (SEM). Unpaired *T*-test, ****p ≤ 0.0001, NS, p = 0.6238. A total of 6–7 pools of 15 mosquitoes each per condition.

The online version of this article includes the following figure supplement(s) for figure 6:

**Figure supplement 1.** *Plasmodium berghei* infection increases megacyte association to the midgut basal surface.

**Figure supplement 2.** Regular granulocytes are not associated with the midgut basal lamina in response to *P. berghei* invasion in *Cactus*-silenced mosquitoes.

(*Figure 6B*). This indicates that *Toll* signaling, through *Rel1*, mediates megacyte differentiation in *dsCactus* females.

We next investigated the effect of *Toll* signaling on the immune response to *Plasmodium* of *dsCactus*-silenced females and megacyte HdMv release. *Cactus* silencing drastically reduced oocyst numbers (median = 0, p < 0.0001, analysis of variance [ANOVA], Dunn's multiple comparison test) relative to *LacZ* controls (median = 127) (*Figure 6C*), and co-silencing *Cactus* and *Rel1* significantly increase oocyst numbers (median = 24, p = 0.0002, ANOVA, Dunn's multiple comparison test) (*Figure 6C*), in agreement with previous reports (*Frolet et al., 2006*). A strong increase in *TM7318* mRNA associated with the midgut 24 hr post-infection was detected in *dsCactus* infected females (13-fold increase), relative to *dsLacZ* (p < 0.0001, Mann–Whitney test) (*Figure 6D* and *Figure 6—figure supplement 1*), indicative of midgut recruitment of megacytes. Furthermore, when *Rel1* was co-silenced with *Cactus*, the levels of midgut-associated *TM7318* mRNA decreased relative to the *Cactus* group and presented a nonsignificant increase in relation to *dsLacZ* controls. In contrast, mRNA levels of *FBN11228*, a marker of regular granulocytes, did not change significantly in *dsCactus* females or after co-silencing *Rel1* and *Cactus*, relative to *dsLacZ* (*Figure 6—figure supplement 2*).

## Discussion

We recently described specific subsets of mosquito granulocytes based on single-cell transcriptomic analysis (*Raddi et al., 2020*). Here, we present a functional characterization of megacytes, a newly described subpopulation of final effector granulocytes and provide direct evidence of their recruitment to the basal surface of the mosquito midgut and their participating in the mosquito immune response to ookinete midgut invasion. The almost complete elimination of *P. berghei* parasites by the mosquito complement system when *Cactus* is silenced was documented more than 15 years ago (*Frolet et al., 2006*). However, the mechanism by which *Cactus* silencing enhanced hemocyte responses to *Plasmodium* infection remained a mystery.

Our transcriptomic analysis indicated that *Cactus* silencing increased the proportion of circulating megacytes, at the expense of regular granulocytes (*Figure 2*). This was confirmed by morphological analysis, ISH, cell counts, and mRNA quantitation of hemocyte-specific markers, *TM7318* (megacytes) and *FBN11228* (regular granulocytes). We also provide direct evidence that, besides being larger, megacytes also have higher plasticity, as they can greatly extend their cytoplasm and flatten their nucleus as they spread on a glass surface (*Figure 1C*).

The lack of DNA replication and the concomitant reduction in the proportion of regular granulocytes indicates that circulating megacytes increase in response to *Cactus* silencing by promoting final differentiation of granulocytes to the megacyte lineage. Besides the dramatic increase in circulating megacytes, *Cactus* silencing also results in megacytes that are even larger and more plastic than megacytes from *dsLacZ* controls. Fine ultrastructural analysis revealed that the cytoplasm of megacytes exhibits extensive large vacuolar structures filled with amorphous material, as well as small vesicles and mitochondria-like structures that are secreted from the cell membrane. In vertebrates, mitochondrial extrusion has been recently documented as a trigger of inflammation. Activated platelets release their mitochondria, both within microparticles or as free organelles; and secreted phospholipase A2 IIA can hydrolyze the membrane, releasing inflammatory mediators, such as lysophospholipids, fatty acids, and mitochondrial DNA, that promote leukocyte activation. Furthermore, extracellular mitochondria also interact directly with neutrophils in vivo, and trigger their adhesion to the endothelial wall (*Boudreau et al., 2014*). Activated monocytes release mitochondria, and their proinflammatory effect on endothelial cells is determined by the activation status of the monocytes that released them. It has been proposed that free mitochondria could be important mediators of cardiovascular disease by inducing activation of type I IFN and TNF signaling (*Puhm et al., 2019*).

Large numbers of megacytes were recruited to the midgut of *Cactus*-silenced females in response to bacterial feeding, forming extensive clusters of cells in close contact with each other, indicating that *Cactus* silencing also results in functional differences in megacytes. Expression of midgut-associated markers of specific hemocyte subpopulations indicates that *Plasmodium* midgut invasion triggers strong recruitment of megacytes to the basal surface of the midgut in *dsCactus* females, in agreement with the documented increase in HdMv associated with epithelial cells invaded by ookinetes (*Castillo et al., 2017*). We also show that co-silencing the transcription factor *Rel1* and *Cactus* disrupts the differentiation of granulocyte to the megacyte lineage observed when only *Cactus* is

silenced, indicating that megacyte differentiation requires a functional *Toll* pathway. Co-silencing *Rel1* and *Cactus* also reduced midgut recruitment of megacytes 24 hr post-infection, a critical time when ookinetes are invading the mosquito midgut, and significantly increases *Plasmodium* survival, relative to *dsCactus* females. We propose that *Toll* signaling promotes hemocyte differentiation into the megacyte lineage, resulting in a dramatic increase in the proportion of circulating megacytes and enhanced midgut megacyte recruitment. This response could mediate the documented increase in HdMv in *Cactus*-silenced females, as HdMv release is necessary for effective activation of mosquito complement (*Castillo et al., 2017*), which ultimately eliminates *P. berghei* ookinetes. The release of free mitochondria-like structures by megacytes from *Cactus*-silenced females raises the question of whether this is an ancient systemic danger signal that promotes immune activation.

## Materials and methods

### Mosquitoes and mouse feeding

*A. gambiae* mosquitoes (G3 strain – CDC) were reared at 28°C, 80% humidity under a 12 hr light/dark cycle and kept with 10% Karo syrup solution during adult stages. For mosquito infections with *P. berghei*, we used the transgenic GFP *P. berghei* parasites (ANKA 2.34 strain) kept by serial passages into 3- to 4-week-old female BALB/c mice (Charles River, Wilmington, MA) starting from frozen stocks. Mouse infectivity was evaluated before feeding by parasitemia levels from Giemsa-stained thin blood films and in vitro microgamete exflagellation counting. Briefly, 1 µl of tail blood was mixed with 9 µl of gametocyte activating medium (RPMI 1640 with 25 mM HEPES (4-(2-hydroxyethyl)-1-piperazinee thanesulfonic acid) + 2 mM glutamine, sodium bicarbonate 2 g/l, 100 µM xanthurenic acid, 50 µg/ml hypoxanthine). After 10 min of incubation exflagellations were quantified using a ×40 objective by phase contrast. Four- to five-day-old mosquitoes were fed when mice reached 3–5% parasitemia and 2–3 exflagellation per field. To feed blood-fed control mosquitoes, 3- to 4-week-old uninfected mice were used. Following feeding, both control and infected mosquitoes were maintained at 19°C, 80% humidity, and 12 hr light/dark cycle until the day of dissection.

### Perfused hemocytes live imaging

Three-day-old adult females were injected with Vybrant DiI (1:10 water diluted, Thermo Fisher Scientific, Waltham, MA, USA) on one side of the thorax. The next day, mosquitoes were injected with 69 nl of either *dsCactus* or *dsLacZ* at 3 µg/µl on the other side of the thorax. After 4 days, hemocytes were ready for perfusion or mosquitoes were used for in vivo live imaging as described below. Mosquitoes were cold anesthetized and, using forceps, a small cut was made in the abdomen. Transfer buffer (95% Schneider media + 5% citrate buffer) was injected at the thorax and 10–15 µl of hemolymph was harvested at the cut-site. This was repeated for five to seven mosquitoes and collected in a microcentrifuge tube stored on ice. To stain the plasma membrane of hemocytes we used CellMask green plasma membrane stain stock solution (C37608, Invitrogen, Waltham, MA, USA) and for the nuclei we used the Hoechst 33342 Solution (20 mM) (Thermo Fisher Scientific, Waltham, MA, USA). Two microliters of fluorescent label solution (58 µl $H_2O$ + 1 µl Cell Mask stock + 1 µl Hoechst stock) was added for every 20 µl of perfusion and 100 µl of this mixture was mounted on an ibidi µ-Slide 18 Well Glass Bottom slide. Cells were allowed to settle for 30 min then imaged. Images were taken on a Leica SP5 confocal microscope using a 63 × 1.4 NA oil objective with 405 nm wavelength laser (at 3% transmission) for Hoechst, 488 nm (5%) for Cell Mask, and 561 nm (3%) for DiI. Pinhole was set to 1 AU and frame average was 12. Z intervals of 1–2 µm encompassing the full cell height was taken every 5 min for 2 hr.

### Bacterial artificial feeding

We used a bacterial mixture obtained from the midguts of the *A. gambiae* G3 from our colony (*Barletta et al., 2019*). A preinoculum was set up in LB media from the frozen stocks containing the bacterial mixture and allow to grow overnight at 28°C, 250 rpm in a shaker incubator. At the day of the experiment, the preinoculum was diluted in fresh LB media and allowed to grow for 2 hr in the same condition described above. Briefly, after 2 hr of growth, bacteria were washed with sterile phosphate-buffered saline (PBS) to remove toxins and the concentration of the culture was estimated based on the optical density (OD) of the culture. At 600 nm, 1OD was considered the equivalent of $10^9$ bacteria/

ml. Three- to four-day mosquitos were fed a sterile 10% sucrose solution containing antibiotics (penicillin, 100 U/ml and streptomycin, 100 µg/ml) for 2 days prior the bacterial feeding. Control group was fed with a sterile 10% BSA solution in Hanks' Balanced Salt Solution (HBSS) without calcium and magnesium and the bacteria group was fed with the same solution containing $4 \times 10^9$ bacteria per feeder. Mosquitoes were dissected 6 hr post-feeding for visualization of hemocytes attached to the midgut basal surface.

## Hemocyte collection, morphology staining, and quantification

Hemocytes were collected by perfusion using anticoagulant buffer (60% Schneider medium, 30% citrate buffer, pH 4.5, and 10% fetal bovine serum [FBS]), pH was adjusted to 7–7.2 after mixing all the components. After perfusion, hemocytes were placed in a µ-slide angiogenesis chamber (ibidi GmbH, Gräfelfing, Germany) and were allowed to settle for 15 min. Cells were fixed for an hour at room temperature by adding 16% paraformaldehyde (PFA) solution in anticoagulant buffer to a final concentration of 4%. Following fixation cells were washed with PBS 0.1% Triton and incubated for 30 min at room temperature with 1 U of phalloidin (Alexa Fluor 488, Molecular Probes, Thermo Fisher Scientific, Waltham, MA, USA) and 20 µM Hoechst 33342 (405, Molecular Probes, Thermo Fisher Scientific, Waltham, MA, USA), both diluted in PBS 0.1% Triton. Cells were then placed in mounting media for storage by adding two drops of Prolong Gold Antifade Mountant (Molecular Probes, Thermo Fisher Scientific, Waltham, MA, USA). For the determination of proportion of megacytes upon *Cactus* silencing, the hemocytes were imaged, the diameter of every cell was measured and classified as granulocytes (cell diameter >12.5–25 µm) or megacytes (cell diameter >25 µm) as mentioned before. The total number of granulocytes and megacytes obtained from hemolymph pooled from 16 to 20 mosquitoes was noted and the percentage of megacytes among granulocytes was determined for each sample. Data from three independent biological replicates were used to plot the graphs.

## Measurement and categorization of the hemocytes by size

The mosquito hemolymph was collected and the hemocytes were allowed to attach on a coated well of 15 µm chamber slide. For each well 8–10 mosquitoes were bled and for every sample, bleeding was done in two wells with a total of 16–20 mosquitoes. Post-attachment, the hemocytes were fixed with 4% *p*-formaldehyde and stained with phalloidin and DAPI (4′,6-diamidino-2-phenylindole) to visualize the morphology. Images were taken for at least 10 random fields for each well and the images were used to measure the cell diameter using Imaris 9.6.0 from Bitplane. Using the 'Pairs' option of 'Measurement points' tool in the software, the largest diameter of every cell was determined. For categorizing the hemocytes into different subtypes, the following size reference was followed for every image analysis. Cells with diameter ranging from 4 to 7.5 µm were classified as prohemocytes, >7.5–12.5 µm as oenocytoids, >12.5–25 µm as granulocytes, and >25 µm as megacytes.

## dsRNA synthesis

Three- to four-day-old female *A. gambiae* females were cold anesthetized and injected with 69 nl of a 3 µg/µl *dsCactus* or *dsLacZ* control. Double-stranded RNA for *Cactus* (AGAP007938) was synthesized by in vitro transcription using the MEGAscript RNAi kit (Ambion, Thermo Fisher Scientific, Waltham, MA, USA). DNA templates were obtained by PCR using *An.gambiae* cDNA extracted from whole body sugar-fed females. A 280-bp fragment was amplified with primers containing T7 promoters (F-TAATACGACTCACTATAGGGTAACACTGCGCTTCATTTGG and R-TAATACGACTCACTATAGGGGCCCTTTTCAATGCTGATGT), using an annealing temperature of 58°C. Double-stranded RNA for *LacZ* was synthetized by amplifying a 218-bp fragment from *LacZ* gene clones into pCRII-TOPO vector using M13 primers to generate a dsRNA control as previously described (*Molina-Cruz et al., 2012*). A 386-bp fragment from *Rel1* gene was amplified using primers containing T7 promoters (F-TAATACGACTCACTATAGGGATCAACAGCACGACGATGAG and R-TAATACGACTCACTATAGGGTCGAAAAAGCGCACCTTAAT) using an annealing temperature of 58°C. For double silencing experiments, 138 nl of dsRNA mixture at 3 µg/µl was injected into female *A. gambiae*.

## RNA extraction and bulk RNAseq library preparation

Hemocytes were collected as previously described above. In short, *A. gambiae* females were perfused using anticoagulant buffer and immediately transferred to a glass tube for attachment. After 1 hr,

hemocytes that did not attach to the glass tube were collected and transferred to a 1.5-ml microcentrifuge containing 800 µl of TRIZOL LS reagent (Invitrogen, Waltham, MA, USA), that correspond to the unbound fraction enriched mainly by prohemocytes and oenocytoids. Hemocytes that attached to the glass surface were washed twice with PBS and resuspended in 1 ml of TRIZOL LS reagent (Invitrogen, Waltham, MA, USA), this corresponds to the bound fraction, mainly enriched by granulocytes. Hemocytes were then lysed in TRIZOL reagent for 15–30 min at room temperature to allow for full dissociation, then stored at 4°C overnight and then at −20°C until RNA extraction. The homogenate of hemocyte samples was transferred to Phase Lock Gel Heavy 2 ml tubes (QuantaBio, Beverly, MA, USA) that had been prespun for 1500 RCF for 1 min, and allowed to incubate for 5 min at room temperature. 100 µL of chloroform (200 µL per 1 ml TRIZOL or TRIZOL plus media) was added, the tubes capped, and then vigorously shaken for 15 s. Samples were then centrifuged for 12,000 RCF, 10 min, 4°C. If the clear, aqueous phase was still mixed with TRIZOL matrix then 100 µl more of chloroform was added, and the samples again mixed vigorously and spun as before. The aqueous phase was then transferred to a fresh 1.5 ml Eppendorf tube and the RNA precipitated by adding 0.25 ml of isopropyl alcohol (500 ml per 1 ml TRIZOL reagent used). 20 µl of glycogen (5 mg/ml) was also added to aid in precipitation and pelleting. Samples were mixed by repeated inversion 10 times, incubated for 10 min at room temperature, and then spun at 12,000 RCF, 10 min, 4°C. All the supernatant was removed, and the RNA pellets washed twice with 75% ethanol (minimum 1 ml of ethanol per 1 ml of TRIZOL used). Each time the samples were mixed by vortexing and centrifuged 7500 RCF, 5 min, 4°C. At the end, the supernatant was removed and samples air-dried until almost dry, but not completely (still translucent). RNA was resuspended with 30 µl of RNAse-free water, pipetting a few times to homogenize and then incubating at 55°C for 10 min to completely resuspend. Samples were then stored at −20°C until library preparation by Bespoke Low-Throughput Team at the Wellcome Sanger institute. Total RNA quantity was assessed on a Bioanalyser and ranged from 300 to 39 ng. mRNA was then isolated with the NEBNext Poly(A) mRNA magnetic isolation module. RNAseq libraries were prepared from mRNA using the NEBNext Ultra II Directional RNA Library Prep Kit for Illumina (New England Biolabs) as by the manufacturer's instructions, except that a proprietary Sanger UDI (Unique Dual Indexes) adapters/primer system was used. Furthermore, Kapa Hifi polymerase rather than NEB Q5 was employed. For bulk RNAseq sequencing samples libraries were run on the Illumina HiSeq 4000 instrument with standard protocols using a 150-cycle kit set to a 75-bp paired-end configuration. Libraries supplied at 2.8 nM and loaded with a loading concentration of 280 pM.

## Bulk RNAseq bioinformatic analysis

Sequencing reads in CRAM format were fed into a personal BASH pipeline to convert cram files to fastq using biobam's bamtofastq program (Version 0.0.191) (*Raddi et al., 2020*). Forward and reverse fastq reads in paired mode were aligned to the *A. gambiae* AgamP4.3 reference genome using hisat2 (Version 2.0.4) and featureCounts (Version 1.5.1) with recommended settings. Count matrices were combined before downstream data processing and analysis within R version 3.5.3 (RStudio version 1.0.153). Downstream normalization, DE analysis, and visualization were done with DESeq2 R package (Version 1.18.1) (*Love et al., 2014*). Base factor was defined as the *LacZ* unbound condition. Data were normalized by making a scaling factor for each sample. First the log(e) of all the expression values were taken, then all rows (genes) were averaged (geometric average). Genes with zero counts in one or more samples were filtered out and the average log value from log (counts) for all genes was subtracted. Finally, the median of the ratios calculated as above for each sample was computed and raised to the e to make the scaling factor. Original read counts were divided by the scaling factor for each sample to get normalized counts. Then, the dispersion for each gene was estimated, and a negative binomial generalized linear model fitted. p values for the DE analysis were adjusted for multiple testing using the Bonferroni correction. Genes were considered as differentially expressed in *Cactus* knockdown compared to *LacZ* control if they had an adjusted p value <0.001 (Wald *T*-test) and a log2 fold change >2. Gene lists with vectorbase IDs were converted to gene annotations with g:Profiler (*Raudvere et al., 2019*). g:Profiler utilizes Ensembl as its primary data source and is anchored to its quarterly release cycle. g:GOSt was used to perform functional enrichment analysis on input gene lists to map the data onto enriched biological processes or pathways. In addition to Ensembl, also KEGG, Reactome, WikiPathways, miRTarBase, and TRANSFAC databases were used. Functional enrichment is evaluated with a cumulative hypergeometric test with g:SCS (Set Counts and Sizes)

multiple testing correction (adjusted p value reported only <0.05). Gene lists were ordered on log fold changes. Complete dataset is available publicly in https://www.ebi.ac.uk/biostudies/arrayexpress/studies/E-MTAB-11252.

## Transmission electron microscopy

Hemocytes were collected by perfusion using anticoagulant buffer, described above and they were allowed to settle on Thermanox coverslips (Ted Pella, Redding, CA) for 15 min at room temperature then fixed 2.5% glutaraldehyde in 0.1 M sodium cacodylate buffer overnight at 4°C, and then post-fixed 1 hr with 1.0% osmium tetroxide/0.8% potassium ferricyanide in 0.1 M sodium cacodylate buffer, washed with buffer then stained with 1% tannic acid in dH$_2$O for 1 hr. After additional buffer washes, the samples were further osmicated with 2% osmium tetroxide in 0.1 M sodium cacodylate for 1 hr. The samples were then washed with dH$_2$O and additionally stained overnight with 1% uranyl acetate at 4°C, dehydrated with a graded ethanol series, and embedded in Spurr's resin. Thin sections were cut with a Leica UC7 ultramicrotome (Buffalo Grove, IL) prior to viewing at 120 kV on a FEI BT Tecnai transmission electron microscope (Thermo Fisher/FEI, Hillsboro, OR). Digital images were acquired with a Gatan Rio camera (Gatan, Pleasanton, CA).

## Mitotracker staining

Hemocytes were perfused with anticoagulant buffer, described above. Cells were incubated at room temperature for 15 min for spreading. Then washed three times with 95% Schneider media, 5% citrate buffer to remove most of the serum from the cells. Hemocytes were placed with 200 nM Deep Red Mitotracker 644/665 which is retained after fixation (Molecular Probes, Thermo Fisher Scientific, Waltham, MA, USA) diluted in 95% Schneider media and 5% citrate buffer. Cells were incubated for 45 min at room temperature in the dark, then washed with PBS and fixed with 4% PFA in PBS for 15 min at room temperature. Hemocytes were then counterstained with phalloidin and Hoechst as described above.

## TM7318 ISH

The ISH protocol includes a permeabilization step with a protease treatment, which compromises the cell morphology. To evaluate the morphology of hemocytes and RNA expression by ISH, we used a two-step protocol to image morphology first and then proceed to image the probes, described in *Raddi et al., 2020*. Hemocytes collected by perfusion 4 days after *dsCactus* injection, fixed and stained with Alexa 488 phalloidin (actin) as described above. Ten random fields of each well were imaged using a tile scan 'mark and find' tool, where coordinates of the field are recorded and can be restored to image the same cells later. Then, hemocytes were subjected to ISH using RNAscope multiplex fluorescent reagent kit v2 assay (cat# 323110, ACDBio, Abingdon, UK) following the manufacturer's instructions. Tyramide Signal Amplification (TSA)-based fluorophores Opal 4- color automation IHC kit (cat # NEL801001KT, PerkinElmer, Waltham, MA, USA) was used for the development of fluorescence (Opal 620 – C3). A specific RNA probe for *TM7318* (cat# 543201-C3; Aga-Transmembrane-C3) designed by ACDBio was used to stain specifically megacytes. At the end of the ISH protocol, hemocytes were placed in prolong gold and reimaged using the 'mark and find' tool to recall the positions of the morphology pictures. Images were merged using Imaris 9.3.1 (Bitplane, Concord, MA, USA). Each well was imaged taking 12 fields per well. Post-imaging, the cell diameter of every cell was measured by phalloidin stain as described previously and the total number of granulocytes were determined for each sample. Among the granulocytes and larger cells (cells with diameter >12.5), the number of cells positive for the *TM7318* probe was counted and their percentage was determined for both the control and *Cactus* silencing.

## Confocal microscopy and tile scan imaging

Confocal images were captured using a Leica TCS SP8 (DM8000) confocal microscope (Leica Microsystems, Wetzlar, Germany) with either a ×40 or a ×63 oil immersion objective equipped with a photomultiplier tube/hybrid detector. Hemocytes were visualized with a white light laser, using 498 nm excitation for Alexa 488 (phalloidin); 588 nm excitation for Opal620 (TM7318 probe) and Vybrant DiI (hemocytes); 644 nm excitation for Deep Red Mitotracker (Mitochondria) and a 405 nm diode laser for nuclei staining (Hoechst 33342). Images were taken using sequential mode and variable z-steps.

For combined morphology and in RNA ISH, we used tile scan 'mark and find' tool included in LASX software to capture the same areas of the slide before and after the hybridization. Image processing and merge were performed using Imaris 9.3.1 (Bitplane, Concord, MA, USA) and Adobe Photoshop CC (Adobe Systems, San Jose, CA, USA).

## RNA extraction, cDNA synthesis, and quantitative PCR analysis

*A. gambiae* hemocytes were collected as described above 4 days after dsRNA injection (*dsLacZ* and *dsCactus*). Hemolymph pools of 20 mosquitoes (5 µl/each mosquito) were placed directly into 800 µl of TRIzol LS reagent (Thermo Fisher Scientific, Waltham, MA, USA). For midgut RNA extraction, pools of 20 midguts were homogenized directly in 1 ml TRIzol reagent. RNA extraction was carried out as described above in the section *RNA extraction and bulk RNAseq library preparation*. Total extracted RNA was resuspended in nuclease-free water and 1 µg was used for cDNA synthesis using the Quantitect reverse transcription kit (Qiagen, Germantown, MD, USA) following the manufacturer's instructions. Quantitative PCR (qPCR) was used to measure *FBN11228* (AGAP011228), *TM7318* (AGAP007318), and *FBN50* (AGAP005848) gene expression in hemocytes cDNA. We used the DyNamo SYBR green qPCR kit (Thermo Fisher Scientific, Waltham, MA, USA) with target-specific primers and the assay ran on a CFX96 Real-Time PCR Detection System (Bio-Rad, Hercules, CA, USA). A 139-bp fragment was amplified for *FBN11228* (F-CCAGCATCGGTACAACGGAA and R-AAGCTCGTGTTTTCGTGCTG). A 150-bp fragment was amplified for *TM7318* (F-AAAACATCCAGAAACACGCC and R-GGATTCCGGTTAAGTCCACC). A 92-bp fragment was amplified for *FBN50* (F-ATCACAAGGTTCCGGCTATG and R-CGTTGGTGTAGGTGAGCAGA). Relative expression was normalized against *A. gambiae* ribosomal protein S7 (*RpS7*) as internal standard and analyzed using the ΔΔ Ct method (ref – Livak and Schmittgen, 2001; Pfaffl, 2001). *RpS7* (AGAP010592) primers sequences were: F-AGAACCAGCAGACCACCATC and R-GCTGCAAACTTCGGCTATTC. Statistical analysis of the fold change was performed using unpaired *T*-test (GraphPad, San Diego, CA, USA). Each independent experiment was performed with three biological replicates (three pools of 20 mosquitoes) for each condition.

## In vivo live imaging

Mosquitoes were prepared the same way for imaging of perfused hemocytes and injected with Vybrant DiI cell labelling (Thermo Fisher Scientific, Waltham, MA, USA) for both *dsCactus* and *dsLacZ*. After 4 days, mosquitoes were starved in the morning and then fed on a BALB/c mouse in the afternoon. Imaging took place the next day at 18–20 hr post-bloodmeal. Mosquitoes were imaged as previously described (*Trisnadi and Barillas-Mury, 2020*). Briefly, 5–10 mosquitoes with legs and head removed were placed between a coverslip and glass slide with craft putty as a spacer. Images were taken on a Leica SP5 confocal microscope using a 40 × 1.25 NA oil objective with 561 nm (3%) for Vybrant DiI. A z-stack with 1 µm intervals was taken to include hemocytes circulating in the hemolymph to the midgut lumen. The z-stack was taken every 1 min for 1–2 hr.

## Visualizing hemocytes attached to the midgut basal lamina

To preserve hemocyte–midgut bound, midguts were quick fixed using a higher concentration of fixative injected straight into the hemolymph of the mosquito (207 nl of 16% PFA). To stain hemocytes, the day before the dsRNA treatment (*dsLacZ* and *dsCactus*), 3- to 4-day-old mosquitoes were injected with 69 nl of a 100 µM solution Vybrant CM-DiI cell labelling solution (Thermo Fisher Scientific, Waltham, MA, USA), final concentration in the hemolymph (approximately 3.5 µM). Engorged mosquitoes fed with 10% BSA solution containing bacteria were anesthetized and injected with 207 nl of 16% PFA, rested 40 s before midgut dissection in 4% PFA solution. After dissected, midguts were placed in ice-cold PBS and opened longitudinally, and the bolus was removed. Clean opened tissues were then fixed overnight at 4°C in 4% PFA. The following day, midguts were washed twice with PBS, blocked for 40 min with PBS containing 1% BSA and washed twice with the same solution. For actin and nuclei staining, midguts were incubated for 30 min at room temperature with 1 U of phalloidin (Alexa Fluor 488, Molecular Probes, Waltham, MA, USA) and 20 µM Hoechst 33342 (405, Molecular Probes, Waltham, MA, USA), both diluted in PBS. Tissues were mounted in microscope slides using Prolong Gold Antifade mounting media (Molecular Probes, Waltham, MA, USA). Hemocytes were

visualized by confocal microscopy and the number of hemocytes per midgut in each biological condition was also analyzed.

## PHH3⁺ + BrdU staining

For BrdU staining, *A. gambiae* females injected with dsRNA were treated for 3 days with a sugar solution containing 1 mg/ml bromodeoxyuridine (Sigma-Aldrich, St. Louis, MO, USA). At day 4 post-dsRNA injection, hemocytes were collected with anticoagulant buffer (70% Schneider media, 30% citrate buffer, and 10% FBS) pH 7.4. Hemocytes were allowed to settle on an ibidi μ-Slide 18 Well Glass Bottom slide (Gräfelfing, Germany) for 15 min at room temperature and then fixed for 30 min with 4% PFA followed by a permeabilization step with PBS 0.5% Triton for 20 min. Hemocytes were washed twice with PBS with 1% BSA before treatment with 2 N HCl for 40 min to denature the DNA. Cells were neutralized with 0.1 M sodium borate (pH 8.5) for 3 min, washed four times with PBS and then blocked with PBS 2% BSA for 1 hr at room temperature. Cells were then incubated with murine anti-BrdU antibody monoclonal (1:100; Invitrogen, MoBU-1, stock 0.1 mg/ml) in blocking buffer overnight in the cold room. Next day, hemocytes were washed twice with PBS 0.1% Tween 20 and incubated with Alexa 594 Goat-anti-mouse (1:2000) in blocking buffer for 2 hr at room temperature. Hemocytes were washed three times with blocking buffer and counterstained with 20 μM Hoechst 33342 (405, Molecular Probes, Thermo Fisher Scientific, Waltham, MA, USA) and then mounted by adding two drops of Prolong Gold Antifade Mountant (Molecular Probes, Thermo Fisher Scientific, Waltham, MA, USA).

For PHH3 staining, hemocytes were collected and fixed as described above. Following fixation, hemocytes were washed three times with PBS 0.1% triton and then blocked with PBS 2% BSA and 20% goat serum for 1–2 hr at room temperature. Hemocytes were then incubated with Anti-phospho-Histone H3 (Ser10) Antibody, Mitosis Marker (1:500, Millipore Sigma, # 06-570) in blocking buffer overnight in a cold room. Next day, hemocytes were washed with blocking buffer three times and then placed in a solution containing Alexa 594 goat anti-rabbit (1:2000) in blocking buffer for 2 hr at room temperature. Hemocytes were washed three times with PBS 0.1% triton and then counterstained with 1 U of phalloidin (Alexa Fluor 488, Molecular Probes, Thermo Fisher Scientific, Waltham, MA, USA) and 20 μM Hoechst 33342 (405, Molecular Probes, Thermo Fisher Scientific, Waltham, MA, USA), both diluted in PBS 0.1% Triton at room temperature. Cells were then placed in mounting media for storage by adding 2 drops of Prolong Gold Antifade Mountant (Molecular Probes, Thermo Fisher Scientific, Waltham, MA, USA).

## Oocyst counting in the midgut

*P. berghei* infections were evaluated by counting oocyst numbers after feeding on an infected mouse. Infected mosquitoes were kept at 20°C for 10 days after feeding when they were dissected, and their midgut fixed in 4% PFA for 15 min at room temperature. After washing with PBS three times, midguts were mounted in a slide and counted under a fluorescence microscope, where live oocysts were identified by their GFP expression.

## Acknowledgements

This work was supported by the Intramural Research Program of the Division of Intramural Research Z01AI000947, NIAID, National Institutes of Health. We thank Kevin Lee, Yonas Gebremicale, and André Laughinghouse for insectary support, and Asher Kantor for editorial assistance.

## Additional information

### Competing interests

Nathanie Trisnadi: is affiliated with Atropos Therapeutics Inc The author has no financial interests to declare. The other authors declare that no competing interests exist.

### Funding

No external funding was received for this work.

## Author contributions
Ana Beatriz F Barletta, Conceptualization, Formal analysis, Validation, Investigation, Methodology, Writing – original draft, Writing – review and editing; Banhisikha Saha, Data curation, Formal analysis, Investigation, Methodology; Nathanie Trisnadi, Octavio AC Talyuli, Gianmarco Raddi, Data curation, Formal analysis, Validation, Investigation, Methodology; Carolina Barillas-Mury, Conceptualization, Supervision, Funding acquisition, Investigation, Writing – original draft, Project administration, Writing – review and editing

## Author ORCIDs
Ana Beatriz F Barletta ⬥ http://orcid.org/0000-0001-9913-3775
Octavio AC Talyuli ⬥ http://orcid.org/0000-0002-7026-463X
Gianmarco Raddi ⬥ http://orcid.org/0000-0003-1056-5403
Carolina Barillas-Mury ⬥ http://orcid.org/0000-0002-4039-6199

## Ethics
Public Health Service Animal Welfare Assurance #A4149-01 guidelines were followed according to the National Institutes of Health Animal (NIH) Office of Animal Care and Use (OACU). These studies were done according to the NIH animal study protocol (ASP) approved by the NIH Animal Care and User Committee (ACUC), with approval ID ASP-LMVR5.

## Decision letter and Author response
Decision letter https://doi.org/10.7554/eLife.81116.sa1
Author response https://doi.org/10.7554/eLife.81116.sa2

# Additional files

## Supplementary files
• Supplementary file 1. List of upregulated genes in *Cactus*-silenced hemocytes.

• Supplementary file 2. List of downregulated genes in *Cactus*-silenced hemocytes.

• Supplementary file 3. List of *Toll* pathway components and final effectors upregulated in *Cactus*-silenced hemocytes.

• MDAR checklist

## Data availability
Sequencing data have been deposited in ArrayExpress under this link https://www.ebi.ac.uk/biostudies/arrayexpress/studies/E-MTAB-11252.

The following dataset was generated:

| Author(s) | Year | Dataset title | Dataset URL | Database and Identifier |
|---|---|---|---|---|
| Ana Beatriz Barletta Ferreira | 2021 | Bulk RNAseq of Anopheles gambiae hemocytes upon Toll pathway overactivation | https://www.ebi.ac.uk/biostudies/arrayexpress/studies/E-MTAB-11252 | Array Express, E-MTAB-11252 |

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
