## [Editor Report]

This article reveals a mechanism by which Toll pathway activation lead to protection against Plasmodium in Anopheles mosquitoes by promoting the differentiation of hemocytes.

---

## [Decision Letter]

**Decision letter after peer review:**

[Editors’ note: the authors submitted for reconsideration following the decision after peer review. What follows is the decision letter after the first round of review.]

Thank you for submitting the paper "Toll signaling enhances mosquito antiplasmodial immunity by promoting differentiation of hemocytes to the Megacyte lineage" for consideration by *eLife*. Your article has been reviewed by 2 peer reviewers, and the evaluation has been overseen by a Reviewing Editor and a Senior Editor. The reviewers have opted to remain anonymous.

Editorial Comments to the Authors:

We are sorry to say that, after consultation with the reviewers, we have decided that this work will not be considered further for publication by *eLife*. Although there are discrepancies between the two reviewers, the overall feeling is that the claims should be better supported by the data. But both reviewers feel that the work is important and that if the claims can be strengthened the paper would be suitable for *ELife*. I would recommend to look carefully at review 1's suggestions. If you can address the most important critics, we will be happy to consider your article as a new submission and handle it as a revision. If you choose this course of action please submit a separate cover letter detailing the changes you have made.

*Reviewer #1 (Recommendations for the authors):*

In this manuscript, Barletta Ferreira and colleagues focus on the impact of Toll signaling on hemocyte differentiation and its impact on antiplasmodial immunity. This manuscript follows earlier demonstration by the authors that megacytes are a new type of hemocytes that appear during infection. The title of the paper suggests that the authors will put together everything in a biologically relevant context. However, the title of the paper is misleading, as there is no formal demonstration of the antiplasmodial role of this phenomenon, neither is there a clear demonstration of a direct differentiation of hemocytes. Instead, the authors demonstrate that Toll induction results in accumulation of megacytes in the mosquito host. If the message of the paper was fully substantiated, I would be very enthusiast for this manuscript. However, because this study repeats experiments already published, and just present data accumulated using RNAi mediated overactivation of Toll pathway, the message of this paper seems rather incremental. I feel there is the need for a number of additional experiments to tie things together to the level of what the title claims, and this would require extensive work.

Some of the problems I detected are:

1. Contrary to what is let thought, there is no biological relevance demonstrated in this paper. The antiplasmodial part is not covered here. Instead, gain of function toll activity is what is studied. This is interesting per se, but does not convincingly put things in context. I would have preferred loss of functions in the context of plasmodium infection for instance. If the authors want to back up their claim, I think both dependency and sufficiency of Toll need to be tested and this all put together in one coherent story.

2. The demonstration of differentiation upon Toll activation is partial. It relies mostly on the absence of EdU positive cells, leading them to conclude that proliferation is not involved and therefore it is differentiation. I have many problems with that. First EdU can be eliminated from cells, and they do not check whether their late timepoint could be affected by this. We do not have timecourse experiment, so it is hard to say. Also, no pH3 staining is performed. More importantly, there is no clear demonstration of which cells would differentiate into which cells. The data only show a final timepoint, and are based mostly on proportions. We at least need to see hemocyte and megacyte numbers to understand whether the numbers increase, decrease, whether this is compatible with a differentiation without proliferation. Even this would be fairly indirect, but at least would build some strong and convergent arguments. Ideally, could an ex vivo experiment be done to demonstrate hemocytes convert into megacytes for instance?

3. We do not have enough controls for the main experiment which is the Toll induction. Can we have documentation that cactus RNAi leads to Toll and Toll only induction? To what levels? Are these levels biologically relevant? As the paper relies mostly on this RNAi, we need the data. Maybe the authors have these data already in their RNAseq? But earlier timepoints would be important, especially as we do not know the kinetic of the described phenotype.

4. In figure 5, we lack unchallenged controls, which would really help interpret this figure.

*Reviewer #2 (Recommendations for the authors):*

This manuscript describes the effect on hemocyte differentiation of activation of the Toll pathway, which is a known antagonist of Plasmodium ookinetes. RNAseq analysis after silencing of Cactus and DNA replication analysis by bromodeoxyuridine incorporation indicated that the total hemocyte granulocyte population declined while megacytes increased, even though the number of granulocytes did not change, consistent with differentiation of a portion of the granulocytes into megacytes. Several diagnostic measurements and a marker were consistent with a Toll activation effect on increased megacyte size and relative abundance. Toll-activated megacytes can issue filopodia-like structures that may interact with other megacytes, and also appear to generate extracellular vesicles containing mitochondrion-like structures. Hemocytes were attracted to the midgut after a bacterial or malaria bloodmeal, but in Toll-activated mosquitoes megacytes were enriched and they formed clusters.

The work presents important new insight into the mechanism of Toll pathway immunity in Anopheles, specifically that Toll activation leads to the differentiation and maturation of a hemocyte lineage to generate megacytes. The megacytes are further stimulated by activated Toll to bind to the midgut, form clusters potentially linked by filopodia, and extrude mitochondrin-like membrane-bound structures that may function as a pro-inflammatory danger signal.

This is an excellent piece of work that fills in important new details about Toll pathway function and Anopheles immunity against Plasmodium, leading up to the function of the hemocyte-derived microvesicles. In my opinion it is suitable for publication with minor revision.

Scientific revision: The only part of the work that could be improved relates to the interpretation of the work represented by figure 5 E & F. Hemocyte markers were measured 4 h after bacterial treatment, or 26 h after malaria infection. Thus, the two assays are not directly comparable, and it is not possible to interpret whether the Plasmodium response is simply the bacterial response delayed by 22 additonal hours, or whether there is any particular specificity of the response caused by the presence of malaria parasites as compared to bacteria alone. By relative expression, the induction of megacyte binding after bacteria appears to be a log-fold higher than after Plasmodium but displayed equivalent levels of significance in bacteria or malaria treatments, whereas the relative expression of the granulocyte marker appears to be a similar value after bacteria or malaria, but the difference is more significant in the Plasmodium case. Why? Basically, because of the time difference of the assays, it is not possible to clearly interpret whether these results mean that bacteria induce a stronger megacyte binding response, or whether the response had simply decayed by 22 h later. Similarly, does the greater significance of granulocyte binding after malaria mean that this is a malaria-specific response, or would the bacterial treatment display the same outcome at 26 h instead of 4 h after treatment? Responding to this comment does not require new experimental work. At least, the authors should address this point in the interpretation of the result in the text, including whether they believe there is an effect in quantity or quality due to Plasmodium presence over bacteria alone. However, if the authors already have data after bacterial treatment at the same 26 h time point as malaria, it would be useful to present this, possibly as a supplementary figure, because this data would be directly comparable to Plasmodium treatment.

---

## [Author Response]

[Editors’ note: the authors resubmitted a revised version of the paper for consideration. What follows is the authors’ response to the first round of review.]

Reviewer #1 (Recommendations for the authors):In this manuscript, Barletta Ferreira and colleagues focus on the impact of Toll signaling on hemocyte differentiation and its impact on antiplasmodial immunity. This manuscript follows earlier demonstration by the authors that megacytes are a new type of hemocytes that appear during infection. The title of the paper suggests that the authors will put together everything in a biologically relevant context. However, the title of the paper is misleading, as there is no formal demonstration of the antiplasmodial role of this phenomenon, neither is there a clear demonstration of a direct differentiation of hemocytes. Instead, the authors demonstrate that Toll induction results in accumulation of megacytes in the mosquito host. If the message of the paper was fully substantiated, I would be very enthusiast for this manuscript. However, because this study repeats experiments already published, and just present data accumulated using RNAi mediated overactivation of Toll pathway, the message of this paper seems rather incremental. I feel there is the need for a number of additional experiments to tie things together to the level of what the title claims, and this would require extensive work.

We have modified the title of the manuscript to better reflect our main findings and it now reads “Hemocyte Differentiation to the Megacyte Lineage Enhances Mosquito Immunity Against *Plasmodium*”. The authors would like to clarify that we are not repeating experiments already published. Although it has been known for many years that silencing Cactus results in almost complete elimination of *P. berghei* parasites, no one had been able to elucidate the molecular mechanism mediating this response. To our knowledge, this is the first direct evidence that activation of a signaling pathway promotes differentiation of hemocytes to a specific subpopulation of cells and that this greatly enhances antiplasmodial immunity. We have conducted a series of experiments to address each of the concerns raised by the reviewer.

Some of the problems I detected are:1. Contrary to what is let thought, there is no biological relevance demonstrated in this paper. The antiplasmodial part is not covered here. Instead, gain of function toll activity is what is studied. This is interesting per se, but does not convincingly put things in context. I would have preferred loss of functions in the context of plasmodium infection for instance. If the authors want to back up their claim, I think both dependency and sufficiency of Toll need to be tested and this all put together in one coherent story.

Point well taken. We had not presented the effect of co-silencing *Rel1* and *Cactus* on *Plasmodium* infection because it had been previously shown to significantly increase parasite survival, relative to *dsCactus* females. We agree with the reviewer that we had not demonstrated that differentiation of granulocytes to the Megacyte lineage in *dsCactus* females was mediated by activation of *Toll* signaling. In the revised manuscript we show that, while *Cactus* silencing dramatically increased the proportion of 2 megacytes, from 3.5% to 76% (p<0.0001, X test) (Figure 6A and B), cosilencing of *Cactus* and *Rel1* reverts this effect, resulting in a proportion of megacytes (6.6%) not significantly different from *LacZ* controls (Figure 6B). This provides direct evidence that *Toll* signaling, through Rel1, mediates differentiation of granulocytes to the Megacyte lineage in *dsCactus* females.

We also show that the dramatic decrease in oocyst numbers in *dsCactus* females (median=0, p< 0.0001, ANOVA, Dunn’s multiple comparison test) relative to LacZ controls (median=127) (Figure 6C), is also associated with a strong and highly significant increase (13-fold) in TM7318 mRNA associated with the midgut of *dsCactus* infected females, relative to dsLacZ (p=0.0001, Mann-Whitney test) 24 h post-infection (Figure 6D), indicative Megacyte recruitment to the midgut.

Furthermore, co-silencing *Cactus* and *Rel1* significantly increased oocyst numbers (median = 24, p=0.0002, ANOVA, Dunn’s multiple comparison test) relative to *dsCactus* females, and reduced TM7318 mRNA associated with the midgut to levels not significantly different from those of *dsLacZ* controls (Figure 6C). In contrast, FBN11228 mRNA levels, a marker expressed in regular granulocytes, did not increase in significantly in *dsCactus* females and levels after co-silencing *Rel1* and *Cactus* were also not significantly different from *dsLacZ* (Figure S5)*.* We conclude that cosilencing *Rel1* disrupts granulocyte differentiation to the megacyte lineage in *dsCactus* females, decreases the association of Megacytes with the mosquito midgut at 24 h post-infection, a time of peak ookinete midgut invasion, and increases *Plasmodium* survival.

2. The demonstration of differentiation upon Toll activation is partial. It relies mostly on the absence of EdU positive cells, leading them to conclude that proliferation is not involved and therefore it is differentiation. I have many problems with that. First EdU can be eliminated from cells, and they do not check whether their late timepoint could be affected by this. We do not have timecourse experiment, so it is hard to say. Also, no pH3 staining is performed. More importantly, there is no clear demonstration of which cells would differentiate into which cells. The data only show a final timepoint, and are based mostly on proportions. We at least need to see hemocyte and megacyte numbers to understand whether the numbers increase, decrease, whether this is compatible with a differentiation without proliferation. Even this would be fairly indirect, but at least would build some strong and convergent arguments. Ideally, could an ex vivo experiment be done to demonstrate hemocytes convert into megacytes for instance?

We agree with the reviewer that BrdU can be lost over time, and therefore it is difficult to determine whether a cell proliferated earlier and the BrdU got diluted after several cell divisions. As suggested, the proportion of hemocytes undergoing mitosis after *Cactus* silencing was also directly evaluated using phospho-Histone 3 (PHH3) staining. Two days post injection, the proportion of PHH3+ hemocytes that adhered to glass (mostly granulocytes) in *dsCactus* mosquitoes (0.7%, n=896) was small, and not significantly different from *dsLacZ* controls (0.6%, n=718 cells) (Figure S2C). At four days the proportions were also similar, with very few hemocytes positive for PHH3 staining both in *dsLacZ* (0.5%, n=799 cells) and *dsCactus* (0.3%, n=1039) (Figure S2C). Moreover, the few hemocytes that were positive for PHH3 in *dsCactus* mosquitoes did not have the size or morphology characteristic of megacytes (Figure S2C). We conclude that megacytes are not proliferating 2 or 4 days after *dsCactus* silencing.

A total hemocyte count was done to determine if the observed increase in megacytes upon Cactus silencing could be due to an overall increase in number of circulating hemocytes. The total number of circulating hemocytes was not significantly different between *dsLacZ* (mean 17,151/mosquito, n=14) and *dsCactus* mosquitoes (mean 23,477/mosquito, n=14, Mann Whitney T-test) (Figure S2A) 2 days post injection. Overall, the mean number of total hemocyes was lower at 4 days post-silencing, but there was also no significant difference between *dsLacZ* (mean 8,725/mosquito, n=14) and *dsCactus* mosquitoes (mean 10,836 /mosquito, n=14, Mann Whitney T-test) (Figure S2A). There was also no difference in the proportion of total granulocytes at 2 days, *dsLacZ* (3.62%) and *dsCactus* females (3.48%, Mann Whitney T-test), or 4 days post-injection, *dsLacZ* (3.91%) and *dsCactus* females (5.47%, Mann Whitney T-test).

In contrast, a significant increase in the proportion of megacyte was already apparent 2 days post-injection in *Cactus*-silenced mosquitoes (2.08% of all hemocytes, p<0.0001, Mann Whitney T-test), relative to *dsLacZ* controls (0.07%) (Figure S2B); with a corresponding decrease in the proportion of other granulocytes from 3.5% in *dsLacZ* controls to 1.4% in *dsCactus* (Figure S2B). At four days post-injection, the differences were more pronounced, with the proportion of megacytes reaching 3.7% in *Cactus* silenced mosquitoes (p<0.0001, Mann Whitney T-test) relative to *dsLacZ* controls (0.07%), again with a corresponding decrease in other granulocytes from 3.8% in *dsLacZ* to 1.8% in *dsCactus* mosquitoes (Figure S2B). Taken together, these data indicate that, although the total number of hemocytes and the percent of total granulocytes remained unchanged in response to *dsCactus* silencing, the proportion of megacytes increased at the expense of other granulocytes and no proliferation of megacytes was observed, based on both BrdU and PHH3 staining. All our findings indicate that *Cactus* silencing promotes differentiation of granulocytes to the megacyte lineage, and this differentiation requires *Rel1*-mediated activation of the *Toll* pathway.

3. We do not have enough controls for the main experiment which is the Toll induction. Can we have documentation that cactus RNAi leads to Toll and Toll only induction? To what levels? Are these levels biologically relevant? As the paper relies mostly on this RNAi, we need the data. Maybe the authors have these data already in their RNAseq? But earlier timepoints would be important, especially as we do not know the kinetic of the described phenotype.

Point well taken. It is not possible to establish whether only the Toll pathway is activated when we silence Cactus. However, the RNAseq analysis revealed that expression of several genes involved in *Toll* signaling or final effector this pathway, such as Toll-like receptors, CLIP proteases, Serpins, C-type Lectins and Defensin are upregulated in *dsCactus* hemocytes (see Table S3 in the revised manuscript). Furthermore, co-silencing Rel1 and Cactus prevented the dramatic increase in the proportion of megacytes that is observed when Cactus alone is silenced, indicating that the differentiation response requires activation of *Toll* signaling through Rel1.

4. In figure 5, we lack unchallenged controls, which would really help interpret this figure.

We have previously shown that bacterial feeding significantly increases granulocytes recruitment to the midgut surface of wild type mosquitoes (that have not been injected with dsRNA) (Barletta et al. 2019, *iScience*). In this manuscript (Figure 5), we evaluated whether there was a difference in hemocyte midgut recruitment between *dsLacZ* and *dsCactus* females. We observed a qualitative difference, as individual or pairs of hemocytes were observed in *dsLacZ*, while hemocytes formed large clusters in *dsCactus* females (Figure 5 A-D). Using markers for specific hemocyte subpopulations we also show that there is massive recruitment of megacytes in *dsCactus* females, with a 250-fold increase in midgut-associated TM7318 mRNA levels, a megacyte-specific marker, relative to *dsLacZ* controls (Figure 5E).

Reviewer #2 (Recommendations for the authors):This is an excellent piece of work that fills in important new details about Toll pathway function and Anopheles immunity against Plasmodium, leading up to the function of the hemocyte-derived microvesicles. In my opinion it is suitable for publication with minor revision.

We thank the reviewer for his insight on the contribution of our work.

Scientific revision:The only part of the work that could be improved relates to the interpretation of the work represented by figure 5 E & F. Hemocyte markers were measured 4 h after bacterial treatment, or 26 h after malaria infection. Thus, the two assays are not directly comparable, and it is not possible to interpret whether the Plasmodium response is simply the bacterial response delayed by 22 additonal hours, or whether there is any particular specificity of the response caused by the presence of malaria parasites as compared to bacteria alone. By relative expression, the induction of megacyte binding after bacteria appears to be a log-fold higher than after Plasmodium but displayed equivalent levels of significance in bacteria or malaria treatments, whereas the relative expression of the granulocyte marker appears to be a similar value after bacteria or malaria, but the difference is more significant in the Plasmodium case. Why? Basically, because of the time difference of the assays, it is not possible to clearly interpret whether these results mean that bacteria induce a stronger megacyte binding response, or whether the response had simply decayed by 22 h later. Similarly, does the greater significance of granulocyte binding after malaria mean that this is a malaria-specific response, or would the bacterial treatment display the same outcome at 26 h instead of 4 h after treatment? Responding to this comment does not require new experimental work. At least, the authors should address this point in the interpretation of the result in the text, including whether they believe there is an effect in quantity or quality due to Plasmodium presence over bacteria alone. However, if the authors already have data after bacterial treatment at the same 26 h time point as malaria, it would be useful to present this, possibly as a supplementary figure, because this data would be directly comparable to Plasmodium treatment.

Point well taken. The blood meal ingested by a mosquito is surrounded by the peritrophic matrix, and bacteria in the gut microbiota proliferate extensively when more nutrients become available as the meal undergoes digestion. The peak of ookinete midgut invasion occurs between 24-26h post-feeding. As the invade the midgut, ookinetes break the barriers, such as the peritrophic matrix, that normally prevent direct contact between the gut microbiota and epithelial cells. As a result, midgut epithelial cells come in direct contact with bacteria from the microbiota and with the immune elicitors they release, and this triggers prostaglandin (PGE2) release by the midgut. PGE2, in turn, attracts hemocytes to the midgut basal lamina (Barletta et al. 2019). Thus, the response of hemocytes to ookinete invasion requires both the presence of bacteria in the bolus and a breach of barriers by the invading parasites. If hemocytes come in contact with a nitrated surface, they undergo apoptosis and release microvesicles in close proximity to the invaded cells. Fortunately, we can still detect hemocyte mRNAs markers of specific hemocyte subpopulations in the cell remnants that remain associated with the midgut after hemocytes undergo apoptosis. In Figure 6 of the revised manuscript we show that midgut-associated mRNA levels of TM7318, a megacyte marker, increase significantly in *dsCactus* hemocytes, and goes back to levels similar to dsLacZ controls when the transcription factor Rel1 is cosilenced with Cactus. We have previously shown that hemocytes are also attracted to the midgut in the dsLacZ control, but here we show that silencing *Cactus* greatly enhances the recruitment of megacytes.

We have previously shown that bacterial feeding significantly increases granulocytes recruitment to the midgut surface of wild-type mosquitoes (that have not been injected with dsRNA) (Barletta et al. 2019, *iScience*).

In these experiments, instead of a sterile blood meal, females are fed a BSA meal containing bacteria from the gut microbiota that were cultured in vitro. As a result, bacteria immediately come in direct contact with epithelial cells, because the peritrophic matrix has not formed yet. This treatment triggers strong PGE2 release and circulating hemocytes are attracted to the basal surface of the midgut. We analyze the midgut 4 hours after feeding and we can observe intact hemocytes, because there has been no midgut invasion and the basal lamina is not nitrated, so hemocytes do not undergo apoptosis. As a result, we can visualize hemocytes directly associated with the basal surface of the midgut and we can easily detect mRNAs from specific hemocytes subpopulations.

When we evaluated whether there was a difference in hemocyte midgut recruitment between *dsLacZ* and *dsCactus* females in response to bacteria (Figure 5). We observed a qualitative difference, as individual or pairs of hemocytes were observed in *dsLacZ* controls, while hemocytes formed large clusters in *dsCactus* females (Figure 5 A-D). Using markers for specific hemocyte subpopulations we also show that there is massive recruitment of megacytes in *dsCactus* females, with a 250-fold increase in midgut associated TM7318 mRNA levels, a megacyte-specific marker, relative to *dsLacZ* controls (Figure 5E).

For clarity, those experiments involving bacterial feeding are now shown separately (Figure 5) from those involving midgut invasion by *P. berghei* ookinetes (Figure 6) in the revised manuscript. The following information has also been added:

“We have shown that the direct contact of bacteria with epithelial cells, before the peritrophic matrix is formed, triggers PGE2 release and attracts hemocytes to the basal surface of the midgut (Barletta et al., 2019). Hemocyte recruitment to the midgut in *dsCactus* females was explored by providing a BSA protein meal containing bacteria. As expected, bacterial feeding attracted hemocytes to the midgut surface in both *dsCactus* and *dsLacZ* control females (Figure 5A and B).”